# Can T Cells Abort SARS-CoV-2 and Other Viral Infections?

**DOI:** 10.3390/ijms24054371

**Published:** 2023-02-22

**Authors:** Leo Swadling, Mala K. Maini

**Affiliations:** Division of Infection and Immunity, Institute of Immunity and Transplantation, University College London, Pears Building, London WC1E 6BT, UK

**Keywords:** SARS-CoV-2, T cell, abortive infection, seronegative, adaptive immunity

## Abstract

Despite the highly infectious nature of the SARS-CoV-2 virus, it is clear that some individuals with potential exposure, or even experimental challenge with the virus, resist developing a detectable infection. While a proportion of seronegative individuals will have completely avoided exposure to the virus, a growing body of evidence suggests a subset of individuals are exposed, but mediate rapid viral clearance before the infection is detected by PCR or seroconversion. This type of “abortive” infection likely represents a dead-end in transmission and precludes the possibility for development of disease. It is, therefore, a desirable outcome on exposure and a setting in which highly effective immunity can be studied. Here, we describe how early sampling of a new pandemic virus using sensitive immunoassays and a novel transcriptomic signature can identify abortive infections. Despite the challenges in identifying abortive infections, we highlight diverse lines of evidence supporting their occurrence. In particular, expansion of virus-specific T cells in seronegative individuals suggests abortive infections occur not only after exposure to SARS-CoV-2, but for other *coronaviridae*, and diverse viral infections of global health importance (e.g., HIV, HCV, HBV). We discuss unanswered questions related to abortive infection, such as: ‘Are we just missing antibodies? Are T cells an epiphenomenon? What is the influence of the dose of viral inoculum?’ Finally, we argue for a refinement of the current paradigm that T cells are only involved in clearing established infection; instead, we emphasise the importance of considering their role in terminating early viral replication by studying abortive infections.

## 1. Introduction

The current dogma states that antibodies, in particular those that neutralise virions, are responsible for blocking infections, whereas T cells are predominantly responsible for clearing established infections. Neutralising antibodies can mediate true sterilising immunity, preventing viruses from entering a cell, whereas T cells require viral entry, antigen processing, and presentation on MHC to recognise an ongoing infection. Here, we challenge the dogma that T cells are less relevant in early viral control. We consider the evidence that *T cells can abort infections*, clearing virus before it reaches the limit of detection by routine assays, resulting in boosted T cell immunity without seroconversion.

A strong, broad and multifunctional T cell response has been linked to protection from severe disease in acute-resolving infections, such as SARS-CoV-2, and contributes to protection (alongside humoral immunity) after vaccination (reviewed in [1,2,3,4,5]). Here, we will instead discuss key emerging questions related to the role of T cells in protection from *overt*/detectable infection. We will first address the possibility that seronegative infections could simply reflect a failure to detect the antibody response, then whether T cell responses are epiphenomena rather than mediators of viral control, and finally consider the role of viral inoculum.

We wish to distinguish abortive seronegative infection (defined here as: subjects in whom viral replication and systemic antibodies remain undetectable) from a number of other distinct outcomes of viral exposure (Figure 1). These include: exposed individuals remaining uninfected due to (a) sterilising immunity, or (b) complete genetic resistance. Genetic resistance can be, for instance, due to a lack of expression of a key viral receptor [6,7,8] (exemplified by CCR5-individuals resistant to HIV [6], and low ACE-2 expression for SARS-CoV-2 [7]). Abortive infection is clearly distinguishable from asymptomatic infection or controlled chronic infection (e.g., HIV long-term non-progressors), where virus is detectable at some stage of the infection and where systemic antibodies are often generated. We would also like to distinguish it from infection with non-replicative virus (defective virions or antigen alone), occult or serosilent infection and late seroconversion (where viral replication is detectable), and finally from an abortive viral life cycle within an individual infected cell [9] (Figure 1).

It is clear from the severity of the SARS-CoV-2 pandemic that the number of hosts completely resistant to infection is relatively low and may be becoming vanishingly rare as variants of concern emerge with greater capacity for transmission or immune escape [10]. However, a degree of natural resistance to overt infection (detectable by conventional laboratory tests) does occur; by studying this we can *hone-in on the type of immunity that is most effective* at early control of infection. Abortive infection also reflects a missed diagnostic category with likely altered subsequent protection and response to new exposure and vaccination. Recognition of abortive infections may allow us to better model and predict the outcome of the current and future pandemics.

Several lines of evidence suggest that some individuals resist detectable infection. Epidemiological data from outbreak studies (e.g., cruise ships, care homes) highlight seronegative individuals in highly exposed cohorts. Relatively low rates of infection in particular age groups or geographical locations despite similar exposure suggest an enrichment of individuals who can resist detectable infection. With the exception of human challenge, it is not usually possible to confirm exposure to the virus on an individual basis and, therefore, to differentiate between avoidance of exposure, genetic resistance, and abortive infection. In cross-sectional and rare prospective studies of cohorts where infection rates are high, a certain level of exposure is expected. Studies of seronegative individuals in these contexts can help to identify abortive infections. For instance, intensive monitoring throughout pandemic waves of infection can overcome some of these difficulties.

## 2. Characterisation of Abortive Infection by Early Sampling of a New Pandemic Virus

When SARS-CoV-2 emerged as a new human pathogen, it offered the opportunity to study individuals first exposure to a virus, where any prior immunity must be cross-reactive. We were uniquely placed to identify abortive SARS-CoV-2 infections in our prospective health care worker (HCW) study due to rapid recruitment (baseline samples collected in the first week of the UK’s first lockdown, March 2020) and intensive monitoring (made possible due to crowdfunding) [11,12]. We identified a high rate of infection in HCW (21.5%) relative to the general public in London over the first 16 weeks of the pandemic [13,14]. Importantly, we could precisely define a group of HCW who did not have a detectable infection, remaining negative on weekly PCR and on a panel of serological tests (weekly Euroimmun and Roche Spike IgG and Roche nucleoprotein total antibody, and pseudovirus and live virus neutralisation) [13,14].

A common limitation of previous studies is the use of a single assay and/or a single time point to determine if individuals are seronegative. Despite the high sensitivity and specificity of commercial and research serology tests, a single measurement with only one assay cannot accurately identify seronegative infections. For instance, when considering antibodies against spike and nucleoprotein over a 16-week period, as well as functional assays such as pseudovirus neutralisation, we were able to identify only 2.75% of our cohort of PCR + HCW as non-seroconverters, whereas using a single assay this was as much as 11% [14]. Viral genetic material is often only measurable for a few weeks [13,15]; therefore, repeated weekly PCR testing was also used to minimise false negatives. Weekly serology and PCR testing resulted in a very ‘clean’ seronegative group in which to study T cell responses and blood biomarkers of potential infection [11]. 

We first noted that SARS-CoV-2-reactive T cell responses in seronegative HCW after potential exposure were higher in magnitude than in a pre-pandemic cohort (taken before SARS-CoV-2 circulated in humans and therefore from truly unexposed individuals) [14]. This suggested that exposure without detectable infection could expand SARS-CoV-2-specific T cells. What was unique about our study was the availability of pre-exposure samples: baseline samples from the first week of UK lockdown in the HCW cohort and PBMC taken in 2019 in a cohort of medical students re-recruited when exposed through close contacts [11]. In both of these cohorts we were able to use paired samples to show significant in vivo expansion of SARS-CoV-2-reactive T cells after potential exposure without seroconversion, with no change in magnitude of anti-viral T cells to a control pool of Flu, EBV and CMV epitopes. Multiple sensitive assays confirmed the presence of SARS-CoV-2 specific T cells in a subset of seronegative individuals (ELISpot using 400,000 cells per well, short term cell lines, intracellular cytokine staining and MHC multimer staining).

We next looked for a completely independent marker of exposure to SARS-CoV-2 in the form of a blood biomarker of infection. Because of our early recruitment, we had access to samples in the weeks leading up to PCR positivity and throughout the acute phase of infection. A single interferon-stimulated gene, IFI27, was identified as the best discriminator of PCR-detectable infection—performing better than any previously identified combination of genes/signature of respiratory infection [15]. Crucially, IFI27 expression was also selectively increased in seronegative HCW who expanded SARS-CoV-2-specific T cells and not in those who showed no change in T cell response [11]. The IFI27 is a blood biomarker of viral infection, and is not specific to SARS-CoV-2. However, complete concordance between blood biomarker detection and SARS-CoV-2-specific T cell expansion in HCW at a time when there was a lack of other circulating viruses (the first UK lockdown) was highly suggestive of SARS-CoV-2 exposure. For the first time, a blood biomarker of infection and an expansion of SARS-CoV-2-specific T cells could be shown to occur together in a subset of highly exposed HCW, indicative of abortive infection.

When studying the specificity of the immune response, we were interested to note that the proteins dominating the response during abortive infection differed from those targeted during overt infection. T cell responses in seronegative individuals after potential exposure to HIV, HCV, and HBV have previously been shown to preferentially target non-structural proteins; however, seropositive infection is dominated by structural-specific T cells [16,17,18,19,20,21,22]. T cell responses in individuals who had detectable asymptomatic or mild SARS-CoV-2 infection preferentially targeted the structural proteins, spike, nucleoprotein and membrane, as was seen for SARS-CoV [23]. By contrast, expansions of T cells targeting the non-structural replication-transcription complex (RTC) of SARS-CoV-2 were enriched after exposure in a subset of seronegative HCW. 

The RTC region consists of the NSP12 polymerase, its cofactor NSP7, and NSP13 helicase, proteins that are essential for the first steps of the viral life cycle. They are, therefore, the most conserved across SARS-CoV-2 variants and clades, across human coronaviruses, and in fact across the whole *coronaviridae* family [24]. The RTC-specific T cells are enriched in the memory response after abortive infection relative to overt infection. Most importantly, HCW who abort infection were significantly enriched for NSP12-reactive T cells that cross-recognise SARS-CoV-2 *prior to exposure* when compared to individuals who went on to have a detectable infection [11]. T cells targeting the RTC are relatively common in pre-pandemic unexposed samples [11,22,25,26]. Together, these data pointed to a role for pre-existing RTC-specific memory T cells being rapidly recruited to blunt viral replication before the infection could be established, explaining their relative enrichment pre-exposure in HCW who went on to abort infection and their in vivo expansion (Figure 2). We have, therefore, identified pre-existing NSP12-specific T cells as a correlate of protection from detectable infection that can be investigated in other cohorts and validated in animal models and vaccine studies.

## 3. Evidence for Abortive, Seronegative Infection from Other SARS-CoV-2 Studies

Strong corroborative evidence for the ability of some individuals to resist overt infection following exposure to an infectious inoculum of SARS-CoV-2 came from the human challenge study. Almost 50% of challenge participants ‘resisted’ overt infection at the low dose used in Killingley et al., with either no detectable virus or only ‘transient’ low level detection, and no induction of a circulating antibody response to SARS-CoV-2 [27]. Whether a T cell or blood biomarker signature of abortive infection can also be seen in those challenged volunteers who remained PCR and antibody negative remains to be determined.

Defining the contribution of T cells to protection from infection or disease upon viral exposure is complicated by the presence of pre-existing T cell responses that cross-recognise SARS-CoV-2 [22,28,29,30,31,32]. It is difficult to differentiate the T cell response recruited into the antiviral response on viral exposure from T cells that may cross-recognise SARS-CoV-2 peptides in vitro, but that do not contribute to viral clearance in vivo [11,25,33]. Several cross-sectional studies have identified T cell responses after exposure in seronegative individuals that are broader and higher in magnitude than those seen in unexposed individuals, suggestive of abortive infections.

One of the first studies to identify T cell immunity in close contacts of hospitalised cases who remained seronegative was Wang et al. [34]. Using overlapping peptides to structural proteins (spike, membrane, nucleoprotein, envelope), Wang et al., noted that 15.94% and 26.09% of close contacts had detectable ex vivo CD4 and CD8 T cell responses, respectively, while only 3.33% and 6.67% of pre-pandemic controls showed such responses. However, using in vitro T cell expansion, this rose to 57.79% and 14.49% of close-contacts showing SARS-CoV-2-reactive CD4 and CD8 T cells, respectively. Similarly, using a sensitive proliferation assay and a novel cellular lactate assay, Ogbe et al., described T cell reactivity to SARS-CoV-2 structural proteins in seronegative acute medicine doctors (CD4 > CD8) that was significantly broader and higher in magnitude than in pre-pandemic samples [35]. Therefore, by employing highly sensitive assays, the induction of T cell responses can be observed following likely exposure without detectable infection.

Another aspect of the T cell response that has been used to highlight differences between pre-existing immunity and that generated by viral exposure is the breadth of response. Le Bert et al., were the first to demonstrate the presence of T cells reactive to non-structural proteins, including NSP7 and NSP13, in unexposed individuals. This included multi-specific responses detectable after short-term peptide expansion, but also in some individuals, responses that were detectable ex vivo [22]. Sekine et al., identified CD4 and CD8 T cell responses to structural proteins at higher frequency in close contacts and blood donor samples taken in 2020 than blood donor samples taken pre-pandemic [36]. Interestingly, no pre-pandemic samples showed ex vivo response to both nucleoprotein and spike or membrane, but 26/28 close contacts showed this T cell reactivity despite only 9/31 seroconverting; this suggested potential exposure driving multi-specific T cell responses in the absence of circulating antibodies [36].

Most studies lack assessment of T cell immunity prior to exposure in the same individuals; however, a comprehensive study of household contacts was able to assess the early kinetics of T cell immunity after exposure through rapid contract tracing [37]. The IL-2-producing T cells targeting nucleoprotein and an in silico designed pool of epitopes conserved across HCoV were significantly enriched in household contacts that did not have a detectable infection relative to those who seroconverted. The preference for IL2 production over IFNγ in SARS-CoV-2-specific T cells in this cohort may suggest a dominant role for cross-reactive central memory CD4 T cell responses in protection from detectable infection.

Taken together, several studies using sensitive assays and looking in detail at the specificity of the T cell response have identified enhanced T cell immunity in exposed individuals. This is suggestive of abortive infection resulting from an expansion of SARS-CoV-2-specific T cells without seroconversion.

## 4. Seronegative Infections: MERS, SARS-CoV and Beyond

The outcome of any viral exposure ranges from sterilizing host immunity, in which there is elimination before replication, to uncontrolled viral replication (Figure 1). By definition, seronegative infections are resolved without the contribution of systemic antibodies and are usually identified due to the induction of cellular immunity. Although difficult to prove conclusively, there are data suggestive of seronegative abortive infections associated with the induction of T-cell immunity for many viral infections. The relative frequency and importance of this type of outcome will, however, vary. In general, seronegative infections have been understudied due to the complexities of identifying an immune response to viral exposure without using serology.

T cell responses to other coronaviruses have been described In seronegative individuals. Over 50% of a cohort of camel workers tested seropositive for MERS spike antibodies, suggesting a high rate of exposure and infection, but interestingly a further 29% of the seronegative workers showed detectable CD4 and CD8 T cell responses to the structural proteins of MERS (spike, nucleoprotein, membrane and envelope) [38]. In a study of abattoir workers in Nigeria, none were MERS-CoV seropositive; however, 30% had detectable ex vivo CD4 and CD8 T cells responses (with no MERS-reactive T cells found in workers from abattoirs not handling camels or non-abattoir workers) [39].

One of the settings in which a role for T cells in resisting overt infection has been extensively studied is that of highly exposed HIV-1 seronegative individuals (reviewed in [40,41]). In a cohort of sex workers at high risk of HIV-1 infection in Nairobi, despite ~90% becoming infected, a plateau in new infections was observed after ~three years of exposure, which is suggestive of a subset of individuals who resist infection long-term [16]. The HIV-1-specific T cell responses, both CD4 and CD8, are enriched in HIV-1-exposed persistently seronegative (termed HEPS) subjects [42,43] and after single time-point exposure, even with high titre infectious virus [44,45], when compared to unexposed controls. Interestingly, low dose SIV infection in macaques can recapitulate this immunity, generating Th1 CD4 helper responses without Abs, which protected against subsequent higher dose exposure [46]. Virus-specific CD4 and CD8 T cell responses have also been described in seronegative partners of individuals chronically infected with HSV [47], and HBV [48] and in seronegative HCW with occupational exposure to HCV [17] and HBV [18]. For influenza, pre-existing CD4 and CD8 T cell responses have been associated with lower rates of infection [49,50], suggestive of T-cell mediated-aborted infections.

Taken together, although intensive prospective studies in highly exposed cohorts, the gold-standard for identifying abortive infections, are often lacking, there is still compelling evidence that abortive infections driving T cell expansions occur on exposure to many viruses.

## 5. Emerging/Unanswered Questions

### 5.1. Could the Antibody Response Have Been Missed in Seronegative Individuals?

In immunocompetent hosts, it may seem counterintuitive that levels of viral infection sufficient to induce an interferon signal and expand T cell responses would not also trigger the humoral arm of immunity. There are a number of reasons why a humoral response may have been missed in studies reporting seronegative infections; primarily, the reliance on inadequately sensitive tests or timing of samples. In our SARS-CoV-2 study we therefore tested for anti-nucleocapsid and anti-spike binding and neutralising antibodies using a battery of sensitive assays at multiple time points before classifying donors as ‘seronegative’ [11]. Sampling of some donors before likely viral exposure and maximal T cell expansion made it unlikely that we had missed a very transient early antibody response. This was further underscored by their lack of spike-specific memory B cells (measured using bait reagents which we had previously shown were sensitive enough to detect memory B cells even once neutralising antibodies have waned to undetectable levels [51]).

Although antibodies are typically expected to be induced before or around the time of T cells, shortly after the viral ramp-up phase (Figure 2), some studies [52] have described delayed induction relative to T cells; this underscored the need for our prolonged follow-up sampling (weekly for 16 weeks, then six monthly for two years). For example, human challenge studies with influenza revealed unexpectedly early kinetics of virus-specific T cells, peaking a week after viral exposure, outpacing the antibody response which was only detected at the four-week time point [53]. In both this challenge study and a community-based influenza study [54], rapid viral control resulting in only mild infection was attributed to pre-existing cross-reactive T cells, but subjects mostly did have detectable virus and subsequently seroconverted, thus not falling into our definition of abortive infection.

In asymptomatic infections occurring outside the context of a closely monitored pandemic or challenge experiment, the time of exposure may be impossible to pinpoint, making it difficult to exclude a transient antibody response that has waned by the time of testing. This is an inevitable limitation in cross-sectional studies reporting T cell expansion in seronegative viral infections or exposures. For example, studies suggesting viral exposure sufficient to expand HBV-specific T cells in healthcare workers with vaccine-induced anti-HBs antibodies [18], and in sexual partners of infected individuals, could potentially have missed a previous infection only marked by transient anti-HBc antibodies [48]. Similarly, landmark studies identifying T cell responses in HIV-exposed, but seronegative, individuals [40] did not usually have the opportunity to exclude transient viraemia or serological responses by serial testing around the time of exposure. However, in one study, repeated sampling was possible in three individuals monitored after accidental parenteral exposure to HBV and HIV; they became infected with HBV, but showed no detection of HIV viraemia or seroconversion on repeated sampling, yet did expand HIV-polymerase specific T cells [45].

The lack of detectable antibodies in serum does not rule out a compartmentalised mucosal humoral response. Although lung-resident B cells have been described in mice and humans [55,56,57], the antibodies they produce would still generally be expected to recirculate and to be measurable in the periphery at some point upon repetitive sampling. However, if these B cells produced an IgA-dominated response, this might remain localised to the mucosa. For example, salivary IgA antibodies were identified in seronegative sexual partners of HIV+ men and in HIV-1-resistant sex workers, suggestive of compartmentalised protective humoral immunity [58,59]. Similarly, in a small cohort of HCW with mild SARS-CoV-2 infection where tears, saliva and nasal fluids were collected in addition to blood, anti-spike antibodies, mainly of the IgA isotype, were detected in 15% in one or more mucosal sites and not in the serum [60]. Furthermore, three children of SARS-CoV-2 cases who themselves remained PCR negative generated anti-S1 IgA in saliva, but only one had systemic SARS-CoV-2-specific IgG [61].

Another possibility in the donors we documented with seronegative abortive infection, whose T cells were preferentially directed against early RTC proteins, is that they may have mounted an antibody response to these non-structural antigens rather than those measured against spike or nucleoprotein. Whilst antibodies to RTC proteins would not contribute to virus neutralisation, they would confirm that the humoral response to the virus remains partially intact in such individuals. Antibodies to non-structural proteins (NSP), including NSP7, 12 and 13, have been well-described in classical SARS-CoV-2 infection cases [62,63] although they are not routinely measured. It will be interesting to see if they are also detectable in abortive infection cases not mounting anti-spike or anti-nucleoprotein antibodies. Antibodies to NSP13 have been found in pre-pandemic sera, likely due to cross-reactivity resulting from the high homology of this protein across human coronaviruses, with their detection correlating with a better outcome of SARS-CoV-2 [64]. This association might plausibly be due to NSP13 antibodies being a biomarker for donors with pre-existing SARS-CoV-2 cross-reactive T cells.

### 5.2. Could T Cells Be an Epiphenomenon in Abortive Infection?

Due to the nature of human observational studies, it was not possible to definitively determine whether the selective changes in T cells observed in our abortive infection subjects were ‘cause or effect’ or perhaps were just one key component of a multifaceted protective response. We postulated that pre-existing memory RTC-reactive T cells would be able to exert immediate cytotoxicity and other effector functions to eliminate cells expressing proteins from the first stage of the viral lifecycle, thereby terminating infection before fully productive replication took off.

However, an obvious alternative, or additional, explanation for abortive infection is that these subjects may have had an enhanced innate immune response. Cellular innate mediators to consider include neutrophils in the nasal mucosa [65,66], NK cells and innate-like T cells such as MAITs [67,68,69], all of which may contribute to the control of acute respiratory infections such as SARS-CoV-2. Abortive infection could also have been mediated by enhanced cell intrinsic immunity through interferon-dependent or independent pathways. A strong type I interferon response induced in the early stages of SARS-CoV-2 infection has been associated with a good outcome; the interferon inducible gene IFI27 was induced to lower levels in the blood of HCW with abortive rather than overt infection [11,15], but this may not be representative of the prototypic IFN-I response. It remains possible that efficient induction of alternative innate mediators contributed to more efficient infection containment. For example, interferon lambda (λ) can be preferentially induced in tissue mucosa and has been shown to have antiviral potential against SARS-CoV-2 [70].

Interestingly, RIG-I can act in an IFN-I and IFN-λ-independent manner to restrain SARS-CoV-2 replication at the first stage of the viral life cycle by interfering with the activity of the RNA-dependent RNA polymerase through competitive binding to the viral genome [71]. Such RIG-I activity in infected respiratory tract epithelia would allow translation of the ORF1ab polyprotein from genomic positive strand RNA to produce the RTC proteins, whilst restraining viral replication and production of structural proteins. Thus, it is conceivable that the selective expansion of RTC-targeting T cells we observed was an epiphenomenon, reflecting the fact that RIG-I responses had aborted infection at the stage when only these proteins could be presented to T cells. However, the fact that we observed a selective enrichment of polymerase-specific T cells already present *before exposure* is against their expansion being simply a by-product selected by RIG-I immunity [11]. As RTC protein-derived peptides will be presented in the context of RIG-I restrained infection, it is certainly plausible that innate mediators such as a local RIG-I response worked together with RTC-specific T cells to shut down and remove virally infected cells before productive replication began.

### 5.3. What Is the Influence of Variable Viral Inoculum on Outcome?

A key confounder in studies linking outcomes of infection with differential immune responses is the inability to control for viral inoculum. Although all subjects were exposed to the same viral strain in our study of the first wave of the pandemic, it is plausible that abortive infection resulted from a particularly low dose of viral inoculum that was unable to establish a full-blown infection. This would be consistent with the lower induction of IFI27 we observed in abortive seronegative than in PCR-detectable seroconverting cases [11]. However, the expansion of selective SARS-CoV-2 reactive T cells from baseline to 16-week memory indicates there was at least sufficient infection for antigen presentation. Moreover, studies in other infection settings do not always support a direct relationship between low viral inoculum and likelihood of successful infection clearance. For example, a very low hepatitis B viral inoculum in chimpanzees allowed entry ‘under the immunological radar’ resulting in persistent infection, whereas a higher load triggered immune clearance [72].

Human challenge studies, in which a carefully standardised viral dose is administered to all subjects, remove this variable, allowing more controlled comparison of immune correlates of protection. Interestingly, a challenge of 36 healthy unvaccinated volunteers with a standardised low dose (10 TCID_50_) of wild-type SARS-CoV-2 resulted in detectable infection in only half of the cohort [27]; 18 developed robust viral replication with nasal virus peaking at 8.87log_10_ and persisting for 10 days, whereas 16 remained negative on twice daily nasal and throat swab PCRs. This provides convincing evidence that variability in immunity, rather than viral inoculum, can drive very different outcomes; any differences detected in pre-existing and expanding T cells in those individuals with and without infection in such a controlled setting will be particularly informative.

Based on the same rationale as viral dose, differing infectivity of viral variants is clearly a major confounder when ascribing infection outcome to immunological parameters. Only the original Wuhan hu-1 strain was circulating at the time of the first wave recruitment of our healthcare worker cohort. However, some of the individuals who resisted detectable infection at that time went on to get infected with the Omicron variant, emphasising that these individuals did not have a complete genetic resistance to SARS-CoV-2 infection.

In addition to the initial viral exposure, it is possible that the presence or absence of repetitive or sustained exposures may alter infection outcome (Figure 2). An inoculum of SIV below the threshold required for recovery of virus or seroconversion did result in T cell expansion that was associated with protection to a subsequent exposure [46]. In the woodchuck hepatitis model, repeated exposure to small amounts of virus induced T-cell responses without seroconversion or detectable viral replication; however, in this setting this immunity did not protect from subsequent infectious doses of the virus [73]. The induction of T cells without viraemia or seroconversion has been attributed to repetitive exposure to HIV in sex workers and to HBV in healthcare workers with needlestick injuries and sexual partners of chronically infected individuals [17,18,48]. The HIV-specific T cells are detected intermittently in HEPS individuals, possibly due to varying levels of antigen exposure or assay limitations [16,42]. Interestingly, in one study the key epidemiological correlate of late seroconversion in sex workers who had initially resisted infection was reduction in sex work, which has been shown to lead to a reduction in detectable HIV-specific T cell responses [16,20]. T cell responses in children with potential horizontal exposure also tend to be transient [74]. Together, this suggests that low level antigen exposure may be important in maintaining a protective T cell response. It is conceivable that some UK healthcare workers recruited in the first wave of the pandemic, when PPE use was sub-optimal, may have experienced a similar low-dose repetitive exposure predisposing them to abortive infection. Alternatively, they may have had a nidus of low dose infection that took longer to be cleared than a clinically detectable infection, and therefore resulted in more sustained T cell stimulation. This is supported by the more prolonged induction of IFI27 observed in our abortive cohort than the shorter, but higher IFI27 signal in those with classical infection [11,15].

## 6. Conclusions: Translational Relevance of Identifying Abortive Infections

The recognition of abortive seronegative infection as a new addition to the spectrum of outcomes following exposure to SARS-CoV-2 has several important implications and raises many questions to be explored in future studies. Most critically, the immune response mounted in individuals with such efficient shutdown of infection provides correlates of protection that could be deliberately targeted and boosted by future vaccines and immunotherapies. Further in-depth studies are needed to examine the homing, durability, fine specificity, TCR usage and range of antiviral effector functions mediated by T cell responses characterising abortive infection. Human challenge studies with a controlled uniform viral exposure and animal experiments allowing depletion of T cells may help to establish the protective potential of the RTC-specific T cells enriched in abortive SARS-CoV-2 infections in our study. Development of vaccines incorporating these highly conserved and early expressed viral replication proteins will allow the testing of their capacity to complement existing spike antibodies and provide additional protection against emerging variants.

However, there are wider implications from the recognition of seronegative abortive infection as a *bona fide* outcome of SARS-CoV-2 exposure; a number of individuals who would previously have been classified as unexposed were in fact likely to have had an abortive infection that was missed by standard laboratory tests. It is important to stress that such individuals would not be expected to have complete resistance to infection, so could still be fully infected by more infectious variants if their immunity is not boosted by effective vaccines. However, their pre-existing immunity, and the boosting of this resulting from abortive infection, is likely to drive a differential response to subsequent exposure to homologous and heterologous viruses and vaccines. Extending the identification of abortive infection to much larger cohorts will allow assessment of the influence of abortive infection on future infection susceptibility and severity, and on responsiveness to vaccination. Larger studies of abortive infection are also crucial to determine if this outcome is associated with particular class I or II HLA alleles, which would provide further support for a causal role for the T cell responses observed.

The major limitations for studying T-cell immunity are the lack of standardised assays, the need for PBMC samples (often requiring large blood draws for in depth analysis and careful sample processing) and the labour-intensive assays, largely precluding their advancement into the diagnostic setting. Some progress has been made in the development of more high-throughput T-cell assays, in particular using small volumes of whole blood and measuring anti-viral cytokine release or its mRNA in response to viral peptides [75,76,77]. Use of these assays should lead to a greater appreciation for the contribution of T cells to protection from viral infection and associated disease.

Immunology studies have tended to focus on comparing the characteristics of subjects with varying outcomes of full-blown infection, whilst those resisting detectable infection have been much less studied. More widespread recognition of the distinct category of seronegative abortive infection outcome could stimulate further immunology studies in other viral infections to better define the unique features of T cells in these settings. Factoring in the proportion of an exposed population who have aborted rather than avoided infection will also provide useful information for public health planning and modelling of the ongoing and future pandemics. The data we have reviewed argue for a refinement of the immunological paradigm implicating T cells solely in limiting and controlling established infections, highlighting that these T cells can also contribute to termination of viral replication in its earliest stages.

## Figures and Tables

**Figure 1 ijms-24-04371-f001:**
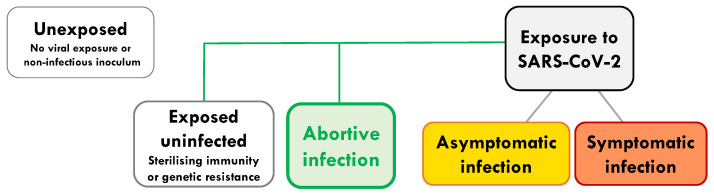
Abortive infection in the spectrum of outcomes from first exposure to SARS-CoV-2. On exposure to SARS-CoV-2 there are a wide range of potential outcomes, such as: exposure without infection (due to sterilising immunity or a cellular genetic resistance); abortive infection, where a low level of infection provides sufficient antigen to expand pre-existing and de novo T cell responses. Abortive SARS-CoV-2 infection can also be identified by raised levels of the interferon stimulated gene, IFI27, in the blood [11]. Abortive infection occurs without induction of systemic antibodies to the virus or sufficient virus to be detectable by PCR. Alternative outcomes are: asymptomatic infection, where the virus and systemic antibodies are detectable in almost all individuals, but no symptoms are induced; symptomatic infection, as with asymptomatic infection, but with measurable symptoms ranging from mild, moderate to severe and fatal infection. Individuals who avoid exposure or have seen only non-infectious inocula, that could not lead to a replicative infection, are considered unexposed.

**Figure 2 ijms-24-04371-f002:**
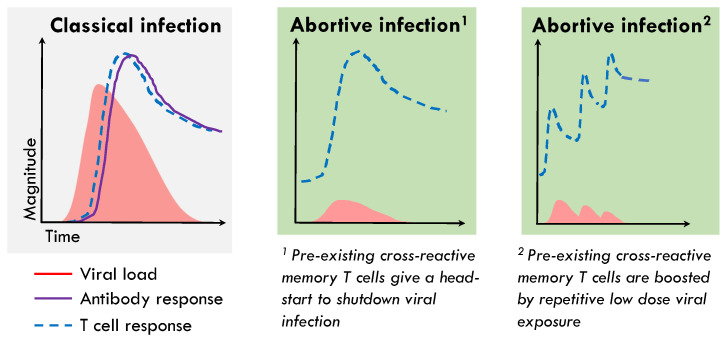
Schema of pre-existing memory T cells aborting infection without seroconversion. During classical acute-resolving virus infections there is a delay between infection and detection of both systemic virus-specific antibodies and T cells, as rare naïve precursors are recruited to the area of infection and draining lymph nodes where they go through several rounds of proliferation, which allows the virus time to exponentially replicate. Due to the association of abortive infection with the presence of pre-existing memory T cells targeting the replication-transcription complex of SARS-CoV-2, we hypothesis that these T cells could be rapidly recruited to, or be present at, the site of infection in the airways and can perform immediate effector functions blunting viral replication before infection is established. This could occur following a single exposure (^1^) or due to low dose repetitive viral exposure (^2^), both of which would result in absence of detectable infection or seroconversion, but the expansion of SARS-CoV-2 specific T cells.

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
