# Peer review of "Can T Cells Abort SARS-CoV-2 and Other Viral Infections?"

_ijms, 2023, doi:10.3390/ijms24054371_

Round 1

Reviewer 1 Report

Hello dear authors,

It is my pleasure to review your review paper titled "Can T cells abort SARS-CoV2 and other viral infections?". I believe that this paper is provide unique perspective on the topic.

The article is really well-written and provides a thorough analysis of the topic.

I have no constructive comments.

Best regards

Author Response

We would like to thank the reviewer for taking the time to review our manuscript and for their kind comments. We are glad that they find the article well written, thorough and a significant contribution to the field. 

We have included a revised manuscript addressing errors recognised during further proofreading and in response to all reviewers. 

Reviewer 2 Report

The authors present a comprehensive overview of the role of T cells in the abortion of SARS-CoV2 and to a lesser degree other viral infections. The manuscript is well written, yet some minor changes in the text and a spell check are required.

In line 9: Some exposed but seronegative individuals will have completely avoided exposure to the virus --> please clarify, if some patients have been exposed, how can they avoid exposure?

In line 16 is a repetition of discuss.

Again a repetition as the following sentences in line 16 and 19 both start with "We discuss"

Line 29, established infectionS plural should be used

Line 71 completely resistant to infection

Line 158 and 159, please clarify: we were interested to note that the proteins dominating the response during overt infection differed from those targeted during overt infection. --> twice overt infection

Line 276, capital L is used for low

Additionally, i would like to request a list of abbreviations, as the audience does not solely have an immunological background in this journal. For example MAIT in line 360 would need explanation for the broader audience.

Author Response

We would like to thank the reviewer for taking the time to comprehensively review our manuscript and for highlighting some oversights and errors. 

We have addressed all comments and provided a point-by-point response below and a version of the manuscript has been uploaded with tracked-changes:

In line 9: Some exposed but seronegative individuals will have completely avoided exposure to the virus --> please clarify, if some patients have been exposed, how can they avoid exposure?

Line 9 has been corrected as follows: "While a proportion of seronegative individuals will have completely avoided exposure to the virus, a growing body of evidence suggests a subset of individuals are exposed but mediate rapid viral clearance before the infection is detected by PCR or seroconversion. "

In line 16 is a repetition of discuss.

Again a repetition as the following sentences in line 16 and 19 both start with "We discuss"

The repetition of 'discuss' has been removed and line 16 re-worded: Despite the challenges in identifying abortive infections, we highlight diverse lines of evidence supporting their occurrence, in particular expansion of virus-specific T cells, not only in SARS-CoV-2 but for other coronaviridae, and diverse viral infections of global health importance (in particular HIV, HCV, HBV).

Line 29, established infectionS plural should be used

S was added to line 29 as suggested. 

Line 71 completely resistant to infection

Resistance has been corrected to resistant as suggested. 

Line 158 and 159, please clarify: we were interested to note that the proteins dominating the response during overt infection differed from those targeted during overt infection. --> twice overt infection

Line 158 has been corrected to indicate that the companion is between abortive and overt infection as such, "When studying the specificity of the immune response in HCW aborting infection we were interested to note that the proteins dominating the response during abortive infection differed from those targeted during overt infection."

Line 276, capital L is used for low

Line 276 has been corrected. 

Additionally, i would like to request a list of abbreviations, as the audience does not solely have an immunological background in this journal. For example MAIT in line 360 would need explanation for the broader audience.

A list of abbreviations has now been included at the end of the manuscript. 

We feel the manuscript has been greatly improved through addressing the reviewers comments. Thank you for taking the time to consider our resubmission and we look forward to hearing from you.

Sincerely,

Professor Mala Maini and Dr Leo Swadling

Reviewer 3 Report

Excellent review. The authors unravel the T-cell function related to abortive infection with a deep discussion, so the current draft is ready for publication. If the authors could increase the font number in the figure.1 ("No viral exposure or non-infectious inoculum", "Sterilising immunity or genetic resistance"), the final version should be better. Thanks.

Author Response

We would like to thank the reviewer for their comments. We have included a new version of figure 1 with increased font size.

Reviewer 4 Report

The article I received for review is excellent in content but needs significant improvement in form. I would first like to praise this work and only then proceed to its critique. This scientific review deals with the very important topic of the T cell response to viral infections. The authors review their own experimental work related to the study of the T-cell immune response to the SARS-CoV-2 virus, in particular the response associated with an abortive viral infection. They also analyze other studies related to variable viral infections for which a large amount of data has been accumulated related to the T-cell responses. The article is logically structured well and well-illustrated. I consider this review timely and important for the development of immunology as a scientific field in general and for understanding the immune response to SARS-CoV-2.

However, there is a huge room for improvement in the presentation of ideas in this review. The text, especially it's the first part, consists of very long sentences that are difficult to understand when reading. In a good text, long sentences are acceptable and appropriate. However, there are too many of them in this text. Each such sentence requires repeated reading and reflection on what the authors wanted to say. In the end, the meaning can be understood, but it takes too much time. I have highlighted too long sentences in the text that are difficult to understand. Although sometimes short sentences need to be improved in order to clarify the meaning. In some cases, I offered my own wording. It is not necessary for the authors to accept them. However, the places where I tried to formulate my own understanding of the stated meanings need to be improved.

I would very much like to advise the authors not to neglect the serious task of improving the text. It is not easy and will require a lot of effort and time. However, without a serious improvement in the text, the work seriously loses.

At the moment, the work looks like an ore with gold, which you have to look at for a long time to see this gold. A good linguistic elaboration of the text should remove the ore and leave only gold, which will not need to be looked at for a long time to see it.

Thus, summarizing the above, I want to say that my criticism relates only to the linguistic form of this review, but not to its factual and logical basis, which are very good.

Author Response

We would like to thank the reviewer for taking the time to comprehensively review our manuscript and for recognising the importance of the manuscript. We have incorporated many of the suggested edits to the text as required to improve the flow and clarity of the text. We have re-edited the whole manuscript thoroughly, shortening many sentences to make it more accessible to readers for whom English is not their mother-tongue.

Reviewer 5 Report

The manuscript presented by L. Swadling and M. K. Maini do not correspond either to the experimental article, since the experimental results are not presented, nor to a review, as it contains multiple references to the results of the authors' research.

I don’t see anything new that the manuscript could give to the reader compared to dozens and hundreds of reviews devoted to the immune response in COVID-19.

Sincerely

Author Response

It is common practice to write a review that refers heavily to a recent article by the authors whilst discussing it in the context of other relevant literature; this is what we have done here. Although there are many existing reviews on COVID-19 immunology they are not addressing the specific topic of abortive infection we have covered here.

Reviewer 6 Report

The basic idea of the review is original. To focus on T cells and to work out their function as responsible for abortive infections. The community is interested in this. Unfortunately, the review has major deficits.

The derivation of the motivation of the review is very lengthy and repetitive. It must be shortened by 1/3 in order not to lose the reader. 

Please discuss: How are the described cross-reactive T cells supposed to eliminate the virus. Especially when the individuals are seronegative? It would be nice if the authors could explain how the early termination of the infection is/ could be mediated by CD4 and CD8 T cells.

There is not enough information in the amount of text, it does not read well, much too long. It is very repetitive. It is constantly repeated that seronegative individuals need to be examined more. But that has been understood.

Usually, the reader learns something new in the very last paragraph that he had not thought of in the review until then.  Now exactly what one has already read several times is repeated.

The citations often do not distinguish between CD4 and CD8. When citing, please also mention the T cell population in question.

The biomarker should be discussed more critically. It probably also occurs in other infections, such as influenza. This could then not be well distinguished from Sars-CoV2.

Minor

It is not usual to write in the text, e.g. Le Bert et a. Cite citations only at the end of the sentence using a number.

Author Response

We would like to thank the reviewer for taking the time to comprehensively review our manuscript and for recognising the importance of the manuscript. We have re-edited the whole manuscript thoroughly to improve the flow and clarity of the text. Additionally, please find a point by point response to your comments below:

The basic idea of the review is original. To focus on T cells and to work out their function as responsible for abortive infections. The community is interested in this. Unfortunately, the review has major deficits.

The derivation of the motivation of the review is very lengthy and repetitive. It must be shortened by 1/3 in order not to lose the reader. 

There is not enough information in the amount of text, it does not read well, much too long. It is very repetitive. It is constantly repeated that seronegative individuals need to be examined more. But that has been understood.

Usually, the reader learns something new in the very last paragraph that he had not thought of in the review until then.  Now exactly what one has already read several times is repeated.

As well as extensive re-writes for clarity, we have shortened the introductory and concluding sections and reduced repetition.

Please discuss: How are the described cross-reactive T cells supposed to eliminate the virus. Especially when the individuals are seronegative? It would be nice if the authors could explain how the early termination of the infection is/ could be mediated by CD4 and CD8 T cells.

We have added a sentence clarifying how memory RTC-specific T cells might eliminate infected cells before productive replication (Line 1294). We have in other sections also highlighted the potential of memory T cells and tissue-resident memory in mediating early control of SARS-CoV-2.

The citations often do not distinguish between CD4 and CD8. When citing, please also mention the T cell population in question.

An indication of whether CD4 and/or CD8 antigen-specific T cell responses were detected in cited literature has been added to the following lines:

Line 740

Line 748

Line 918

Line 921

Line 933

Line 935

The biomarker should be discussed more critically. It probably also occurs in other infections, such as influenza. This could then not be well distinguished from Sars-CoV2.

The section describing the use of the blood biomarker of infection, IFI27, has been re-written to make it clear that it is not SARS-CoV-2 specific.

Line 324… “A single interferon-stimulated gene, IFI27, was identified as the best discriminator of PCR detectable infection - performing better than any previously identified combination of genes/signature of respiratory infection15. Crucially, IFI27 expression was also selectively increased in seronegative HCW who expanded SARS-CoV-2-specific T cells and not in those who showed no change in T cell response11. IFI27 is a blood biomarker of viral infection, and is not specific to SARS-CoV-2. However, complete concordance between blood biomarker detection and SARS-CoV-2-specific T cell expansion in HCW at a time when there was a lack of other circulating viruses (the first UK lockdown) was highly suggestive of SARS-CoV-2 exposure. For the first time, a blood biomarker of infection and an expansion of SARS-CoV-2-specific T cells could be shown to occur together in a subset of highly exposed HCW, indicative of abortive infection.”

Minor

It is not usual to write in the text, e.g. Le Bert et a. Cite citations only at the end of the sentence using a number.

It is actually common practice to highlight a few key manuscripts within the text of reviews by referring to the lead author, rather than just providing a reference at the end of the sentence in the form of a number.

Round 2

Reviewer 5 Report

The manuscript after revision, cannot be called a Review. The authors present their own reflections at many points, the text is rather an essay than a review (with the analysis of the literature).

In its present form, the text hardly corresponds to the reviews usually published in this journal.

Reviewer 6 Report

1. The text could still be shortened to better convey the message. (The basic idea of the review is original. Unfortunately, it does not read well, much too laborious. It needs to be denser on information and crisper.)

2. The role of the T cells is too vague, please be more specific.

3. A cartoon that summarises the many individual parts of the text would be helpful in order not to lose the reader. 

4. Please, consider other ways of terminating an infection such as Loo et al. DOI: 10.1371/journal.pbio.3001967